# Unitary Entities Are the True “Atoms”

**DOI:** 10.3390/e27111119

**Published:** 2025-10-30

**Authors:** Chris Jeynes, Michael Charles Parker

**Affiliations:** 1Independent Researcher, Tredegar NP22 4LP, UK; 2School of Computer Science and Electrical Engineering, University of Essex, Colchester CO4 3SQ, UK

**Keywords:** reductionism, monad, entropy, emergence, geometrical algebra, hyperbolic space

## Abstract

Quantitative Geometrical Thermodynamics (QGT) exploits the entropic Lagrangian–Hamiltonian canonical equations of state as applied to entities obeying the holographic principle and exhibiting Shannon information, the creation of which measures the (validly defined) “entropic purpose” of the system. QGT provides a physical description for what we might consider the true “atoms” of physical science and has also recently enabled a number of significant advances: accounting ab initio for the chirality of DNA and the stability of Buckminsterfullerene; the size of the alpha particle (and other nuclear entities) and the lifetime of the free neutron; and the shape, structure, and stability of the Milky Way galaxy. All these entities, ranging in size over more than 38 orders of magnitude, can each be considered to be an “atom”; in particular, the size of the alpha is calculated from QGT by assuming that the alpha is a “unitary entity” (that is, than which exists no simpler). The surprising conclusion is that clearly compound entities may also be physically treated as unitary (“uncuttable”) according to a principle of scale relativity, where a characteristic size for such an entity must be specified. Since QGT is entropic, and is therefore described using a logarithmic metric (involving hyperbolic space), it is not surprising that the length scale must be specified in order to account for unitary properties and for an entity to be appropriately considered an “atom”. The contribution to physics made by QGT is reviewed in the context of the related work of others.

## 1. Introduction

In this work, we intend to review Quantitative Geometrical Thermodynamics (**QGT**), for it is through QGT that we have developed the general idea of the “unitary entity” (than which nothing simpler exists). “Unitary entity” is not to be confused with the term “unitary operator” (whose adjoint is its inverse, roughly speaking). We use “unitary entity” rather than the (effectively synonymous) “monad” or “atom”—terms which are now rather ambiguous.

An “atom” used to be the name we gave to hypothetically *uncuttable* things. Ludwig Boltzmann based his seminal statistical mechanics on the hypothesis of the reality of such “atoms” in his “*atomic theory of gases*”. Despite the essential correctness of his approach, throughout the 19th century the reality of atoms was doubted, and in 1906 Boltzmann took his own life in despair. This was tragic, since within only a few years the existence of Boltzmann’s atoms was proven beyond doubt by a sequence of fundamental scientific breakthroughs: Jean Perrin published his observational corroboration of Einstein’s 1905 atomic theory of Brownian motion (observed by Robert Brown in 1827; Perrin was recognised for this work with the 1926 Nobel Prize), and Niels Bohr published his “theory of the atom” in 1913 which (amazingly) accounted for the “Balmer lines” of hydrogen (observed by Johann Balmer in 1885 but mysterious until Bohr’s work). Bohr was recognised with the 1922 Nobel Prize for this theory.

Thus, early in the twentieth century it was established that, yes, atoms were real. But reality is elusive; it was also clear that atoms had *structure*: an electronic arrangement surrounding their nuclei. In particular, it was also very clear, very early, that it was remarkably easy to detach electrons from their “atoms”. “Atoms”, oxymoronically, were ***not*** “uncuttable”! In fact, Ernest Rutherford was awarded the 1908 Nobel Prize for demonstrating that the alpha particle was, precisely, the nucleus of the helium atom.

All this is very well known. What is not so well known is that the English “metaphysical poets” of the 17th century took inspiration from the *idea* of (uncuttable) “atoms” promoted by the new microscopes that made the commonplace look extraordinary. Cassandra Gorman (2021, p. 22) [1] observed that these poets “*wrote poetry that meditated on the liberating power of atoms to dissolve and recongregate into renewed and resurrected forms.*” Suddenly the physical world had become much more complex and beautiful than anyone had imagined, as is excellently reviewed by Kevin Killeen (2017) [2] who provides a corrective to our ideas of the emergence of “scientific modernity” which (as he puts it) “*is often still viewed as a sad but necessary putting aside of the poetic*”. But Jeynes et al. (2023) [3] reflect a growing consensus on the reality of the poetry of physics.

In her monograph, Gorman notes that Henry More (1614–1687) originated the neologisms of “***indivisible***” (still in use today) and “***indiscerpible***” (meaning “resists being torn asunder”: now regarded as obsolete). We should comment on the prescience of these 17th century philosophers to distinguish between the “indivisible” (which—in their terms—even God could not divide) and the “indiscerpible” (not impossible to divide, but very difficult).

We usually regard the related “monad” as an important philosophical idea of Leibniz (1714) [4], but More already explained it resonantly in the *Glossary* of his *Philosophicall Poems* (1642) [5]: “*Monad.* Μονάς, *is Unitas, the principle of all numbers, an embleme of the Deity: and the Pythagoreans call it* Θεός, *God. It is from* μένειν *because it is* μόνιμος, *stable and unmovable, a firm Cube of itself. One time one time one still remains one*” (meaning 1^3^ = 1). But Leibniz’ *Monadology* is over half a century later than More’s “*monads*”.

Henry More, together with all his 17th century readers, would have heard in the word “Cube” (derived from the idea of *unity*) echoes of the biblical “Holy of Holies” (a *cube* of dimension “20 cubits”—that is, about 9 metres) well known from *1Kings* 6:20, a text probably significantly earlier than the 6th century BCE. The (deliberate) Christian echo of this is the “Holy City”, a *cube* of 12000 stadia (about 2200 km) in size; see *Revelation* 21:16, a text dated 68 CE by John Robinson (1976) [6] and 68–70 CE by Jonathan Bernier (2022) [7]. Note that More did not think it necessary to state explicitly that *God is One*, although he did mention the Pythagoreans (well known then for supposing the cube to be the source of the element of earth). Frank Wilczek (2021; p. 72) [8] therefore falls into a common fallacy when he attributes the “atomism” of the “scientific revolution” to Democritus: the intellectual power of the idea is predominantly sourced in the contemporaneous breakthroughs in the natural philosophy of the 17th century.

Thus, we ask: Do “*indivisible*” (or at least, “*indiscerpible*”) atoms even exist? Maybe it is the “fundamental particles” of physics that are the (uncuttable) “atoms”? The trouble is that although electrons are regarded by everyone as “fundamental particles” (and everyone agrees that they are “uncuttable”), there is no universal agreement on anything else. Protons and neutrons are made of quarks, and the Standard Model involves a zoo of “fundamental” particles. Nevertheless, we are not sure how correct this model is, given that there remain a number of troubling lacunae (despite its remarkable successes). In particular, the Standard Model does not (yet) include gravity.

In addition, Karen Barad (2007) [9] has cogently questioned whether it is even valid to speak of *individual* particles: since electrons are indistinguishable, we cannot (strictly speaking) single out any individual. Of course, the fiction of “individual” electrons is both convenient and very fruitful: it is “*True Enough*” (Catherine Elgin, 2017) [10] for almost all purposes!

But Parker et al. (2022, [11], see Figure 1) by using the fundamental thermodynamic (entropic) basis of QGT, the holographic principle, and the principle of scale relativity, have shown that treating the alpha particle (in its ground state) as a “unitary entity” enables its matter radius to be correctly determined without any recourse to quantum mechanics. This is startling, since it also undermines the “reductionist” model of scientific progress that has served us so well since Galileo: if the alpha is, properly, an atom (“*indiscerpible*”, in More’s terminology), then “reducing” it to its constituent four nucleons does not give a more “fundamental” description.

This review surveys first the growing wider interest in “geometrical thermodynamics” (Section 2) and then the substantial advances in QGT (Section 3). The QGT formalism may be unfamiliar, so we sketch it in some detail (Section 4) and then briefly summarise (Section 5) what we consider *atoms* to be: *unitary entities*. We try to tie up some loose ends (Section 6) and conclude (Section 7).
Figure 1The ^8^He nucleus is modelled in QGT as a unitary alpha particle with a four-neutron shell which is a holomorphic pair of holomorphic neutron pairs (see Parker et al. 2022 [11]: for image see https://onlinelibrary.wiley.com/toc/15213889/2022/534/2 (accessed on 27 October 2025)). The ^8^He nucleus has an observed matter radius of 2.49 ± 0.04 fm (observations summarised by Angeli 2004 [12]) and a matter radius calculated by QGT as 2.51 fm.
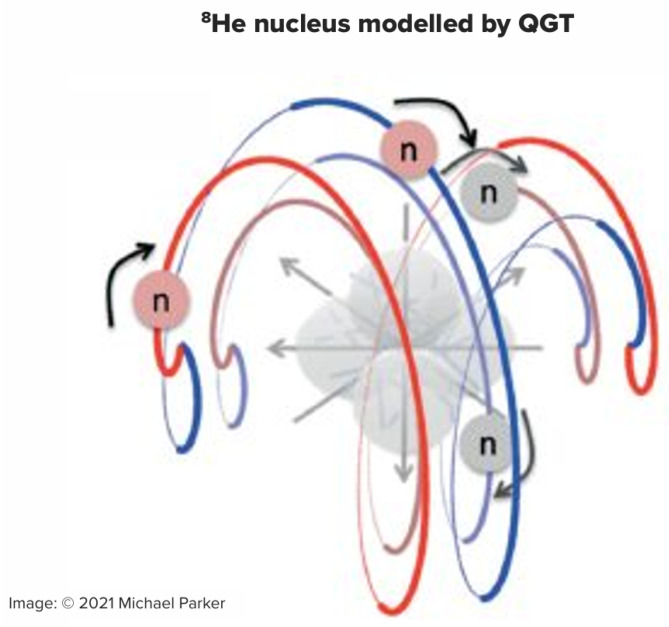


## 2. Geometry in Thermodynamics

The concept of *geometrical* thermodynamics is still relatively unrecognised, although it has roots in the original thermodynamical advances of the 19th century. Mostly, thermodynamics has been considered as a highly quantitative (and successful!) descriptor for use in the analysis of chemical processes, the efficiency of engines and energy transducers of all types (steam, internal combustion, diesel, etc., where matter is transformed from one state into another), and as the source of much philosophical speculation, particularly with respect to the Second Law. In the 20th century, the isomorphism between the theory of information and entropic thermodynamics was recognised; the “efficiency” of computers, telecommunications, and information production was also found to be of fundamental importance. Moreover, the geometry of general relativity and black holes was also discovered to have a profoundly thermodynamic character.

Toikka et al. (2023) [13] have described the *boundary of stability* in thermodynamic systems, making use of the Gibbs–Duhem equation (Willard Gibbs was a giant of 19th century physical chemistry, but until his death in 1916 Pierre Duhem never believed that *atoms* were real). All boundaries are “geometrical” things: the Gibbs–Duhem equation is a fundamental but rather limited relation since it applies only to “homogeneous” thermodynamic systems.

Gibbs himself originally used geometrical diagrams as surface representations of thermodynamic potentials (now known as a “contact geometry”), which served as a bridge between thermodynamics and classical mechanics. His treatment of thermodynamics naturally involves state spaces and conjugate variables, straightforwardly leading to a geometric phase space description (a very good example of which is Carathéodory’s principle). Frank Weinhold has “*called fresh attention to the special beauty and profundity of the work of J. Willard Gibbs*”, crediting him with a “*new representation of equilibrium thermodynamics [using] an intrinsically geometrical structure*” [14].

It was Edwin Wilson who in 1901 popularised the vector calculus [15] that Gibbs and Heaviside derived from Hamilton’s treatment of quaternions that William Clifford had combined with Hermann Grassmann’s treatment of bivectors to generate the Clifford algebra. But it is only quite recently that David Hestenes [16] has advocated the systematic use of geometrical algebra methods to simplify the methods of mathematical physics, and which Parker & Jeynes (2019) [17] have also deployed to underpin QGT.

Thermodynamic stability was also investigated in some detail by Gromov & Toikka (2020) [18] who point out the following: “*The idea of geometric interpretation of thermodynamics, originally suggested by Gibbs at the end of the 19th century, later formulated within the context of contact geometry by Robert Hermann in 1973 … forms a natural framework for a rigorous study of (equilibrium) thermodynamic systems*.” Recent work using contact geometry methods includes Bravetti’s [19] survey of “contact Hamiltonian dynamics”.

Geometrical methods were used by Lu et al. (2023) [20] to investigate (non-equilibrium) quantum thermodynamic effects in interacting quantum dot systems; Córdoba & Isidro (2015) [21] have also shown how complex geometry yields interesting results in fluctuation theory (building on George Ruppeiner’s [22] classical work).

Ramirez & Le Gorrec [23] have given a rather comprehensive formalism to articulate irreversible port-Hamiltonian systems (these are open systems—that is, systems with “ports” open to energy transfers). Port-Hamiltonian systems have attracted much attention, since they are a way of applying a systematic formalism to open interconnected systems (see recently, for example, Lorenz et al. [24]). Note that “port-Hamiltonian” systems are a systematic way to introduce irreversibility: that is, use of the Hamiltonian (or, equivalently, the Lagrangian) does not *require* reversibility.

But note also that Lagrangian–Hamiltonian formulations additionally imply a symplectic geometry framework, which in turn also implies the possibility of a systematic complexification of the formalism (and Parker & Jeynes [25] have now achieved this, including a complexification of time). This is because the symplectic form may be interpreted as a bivector in geometrical algebra (Hestenes [16]); also to be seen in the Riemann–Silberstein description of Maxwell’s equations for electromagnetism [26]. Thus, any treatment of thermodynamical issues resting on a Hamiltonian formalism will always intrinsically possess a *geometrical* interpretation.

Moreover, QGT is based on geometrical algebra methods: Parker & Jeynes [17] use bivectors (and the Clifford algebra) systematically in setting up their formalism, and point out (see their Appendix A) that the isomorphism between the real-valued 4-dimensional (quaternion) vector space and the 2-dimensional complex spinor space is well documented (for example, Pandey & Chakravarti [27] who handle Hestenes’ geometrical algebra very similarly to Parker & Jeynes [17]). Kyriaki Aslani [28] also shows (in her Appendix B) that bivectors are precisely isomorphic to the imaginary unit.

Hans Öttinger [29]’s systematic treatment of “symplectic integrators” applied to (irreversible) thermodynamic systems should therefore also be viewed as geometrical. Unfortunately, symplectic geometry also opens up the conundrum of continuity, which complex analysis presupposes: Roland Riek [30] has claimed to prove that time must necessarily be discrete! It is obvious that the Uncertainty Principle is also incompatible with strict continuity; however, current methods appear to be sufficiently satisfactory so we will ignore these problems, although interestingly, Reguera et al. [31] also summarises very helpfully how to recover discrete behaviours (characterised by the Fokker–Planck equation, see their Equation (57)) using canonical methods.

Raftery et al. [32] have observed a dissipation-induced classical-to-quantum transition in a Jaynes–Cummings dimer model that has been recently thoroughly investigated by Vivek et al. [33]. A Hamiltonian analysis is now standard for modelling Jaynes–Cummings apparatuses (despite the J-C model being dissipative) and Raftery et al. show in detail how the appropriate system description switches from semi-classical to quantum as the dissipation is introduced.

We regard this as an example of geometrical thermodynamics, with the “thermodynamics” being introduced by the dissipation and the (symplectic) “geometry” implied by the Hamiltonian treatment. Indeed, we have shown how these apparently contradictory phenomena of dissipation and Hamiltonian’s reversible symplectic geometry can be coherently united within QGT [17].

Geometrical approaches in thermodynamics up to now have been related to the issues of abstract geometrical structures (e.g., metrics, connections, curvatures, etc.) in state space; such state spaces being described by a set of state variables (e.g., energy, volume, temperature, pressure) each corresponding to a point in the thermodynamic state space of the system under consideration.

The *curvatures* mentioned above are clearly “*geometrical–physical ‘objects’*” as Rubí & Gadomski (2003) [34] point out in a work which explains a valuable “mesoscopic non-equilibrium thermodynamics” (mNET, reviewed by Reguera et al., 2005 [31]) approach to quantifying nucleation and growth (NG) phenomena. mNET is significant because such (NG) phenomena are fundamental to effectively all real processes: reality is non-equilibrium (with entropy always increasing), and we operate at the mesoscopic scale! In the case of mNET, Rubí & Gadomski’s “curvature” refers directly to the (actual average) grain size of the ingredients in NG processes, although they also say that it is probably a “*candidate for the entropic barrier*” in their formalism.

State-space approaches have been particularly described by the Weinhold [35] and Ruppeiner [36] geometries; the former being useful for studying thermodynamic stability (see Andresen et al. [37] for a close discussion of “thermodynamic geometries”), and the latter for phase transitions and critical points. Guo & Miao [38] have recently helpfully described these geometries in the context of certain representations of black holes, but pointed out that the Weinhold and Ruppeiner geometries are essentially equivalent.

On the other hand, the QGT of Parker and others defines a set of variables with thermodynamic characteristics directly in real space–time (not only an abstract phase space), with space–time properties that are geometrically related to each other (that is, also involving derivatives and transformations from Euclidean into hyperbolic space) so that QGT analyses the actual space–time geometry of a system and its thermodynamic evolution (including its stability) according to the relationships between conjugate pairs of the variables in a complexified symplectic phase space. 

This is in contrast with the earlier Weinhold and Ruppeiner approaches that use not symplectic but Riemannian geometries (and Euclidean spaces). Moreover, Parker and others now extend these earlier equilibrium thermodynamics treatments into a more analytical (symplectic) Hamiltonian/Lagrangian one, which being naturally complexified also unifies the handling of both reversible and irreversible processes (PJ23a [25]). However, they (PW25a [39]) also explicitly exploit the information-theoretic aspects and geometries that have always been implicit (PJ19 [17]).

## 3. Geometrical Thermodynamics and Holomorphism

Quantitative Geometrical Thermodynamics (**QGT**) was introduced in 2019 by Parker & Jeynes [17], building on previous work by Parker & Walker (2010) [40]. Figure 2 indicates the approach taken: *information* is represented by meromorphic functions (which are holomorphic nearly everywhere, with isolated points of non-analyticity in space–time) and the basis vectors of *entropy* and *information* are Hodge duals. *Info-entropy* as a bivector is constructed using a QGT holomorphic function across space–time that also allows the construction of an “entropic Hamiltonian” satisfying the canonical entropic equations of state (described in a hyperbolic or logarithmic space; as might be expected from their thermodynamic nature), as well the variational entropic Euler–Lagrange equations based on the appropriate entropic Lagrangian, which also employs the Boltzmann constant *k_B_* as its fundamental quantum unit.

An idea critically important for QGT is *holomorphism*, which is a mathematical description for a function that allows analytic continuation (and holomorphic functions are the main objects of interest in complex analysis). A function being “holomorphic” (literally meaning “*the shape of wholeness*”) is a very strong mathematical condition (involving the Cauchy–Riemann equations), so that QGT being cast in terms of (intrinsically complex) holomorphic (or meromorphic) functions is already a suggestive move.

Physically, holomorphism can be thought to possess the property of “redundancy” since the function’s value *here* may also be perfectly calculated from its value *there*; that is to say, the function’s value at any location may also be found by calculations at numerous alternative locations. Mechanical structures are always designed redundantly for stability, resilience, and toughness, so that performance is not dependent on a single location but is multiply distributed. Thus, a holomorphic (unitary) physical structure is necessarily stable. In QGT, we make explicit this *physical* holomorphism and show how it also entails a maximum-entropy (or stable) state.

We should add that this holomorphic redundancy in QGT also expresses non-local effects (such as the non-local constraints represented by the variational principles). It is well known that quantum mechanics is a non-local theory to which microscopical effects must conform: the current theory of general relativity (accounting for cosmic phenomena) is a local one. QGT applies at all scales.

The double logarithmic spiral (**DLS**) exhibiting dissipation is proved to be a fundamental eigenfunction of the entropic Hamiltonian, and the thermodynamically reversible double helix is a special (non-dissipating) case of the DLS. Both spirals are holomorphic, in the sense that each is composed of a pair of mutually C_2_ symmetry structures (using the crystallographic notation), which spatially combine together so that a geometrical algebra description has them mathematically isomorphic to a Riemann–Silberstein bivector.

To show that these mathematical manipulations touch reality adequately, the energy required to transform one form of DNA to another was calculated by QGT (see Figure 1 of PJ19 [17] *passim*), reproducing the experimental results of Bryant et al. [41]. The observed virial mass of the Milky Way galaxy was also approximately calculated based on a simplified DLS model (see Figure 2 of PJ19 [17] *passim*).

Table 1 of Parker & Jeynes, 2019 (PJ19, [17]), shows a formal isomorphism between conventional kinematical quantities (based on Planck’s constant *h*) and their entropic equivalents (based on Boltzmann’s constant *k_B_*), notably between the classical action and the “Exertion” (proved separately to be proportional to Edwin Jaynes’ “Caliber” [42]). In particular, where *action* is a line integral over the appropriate Lagrangian (in classical or quantum mechanics), *exertion* is simply the corresponding line integral over the *entropic* Lagrangian (and is thereby a thermodynamic quantity).

It is curious that QGT shows that *thermodynamics* may determine the shape (or, equivalently, the structural stability) of entities. For example, DNA is stable, and yet the energy required to deform one form of DNA to another is correctly calculated ab initio by QGT (see Figure 1 of PJ19 [17]). Moreover, using QGT, one can prove that *right*-handed DNA is the thermodynamically favoured form according to the Second Law (as is observed, see Appendix A and Equation (A.17) of PJ19 [17], which gives a more rigorous and explicit argument than that in Parker & Walker 2010 [40]). Parker & Jeynes (2020) [43] have also shown that QGT explains the stability of Buckminsterfullerene (see Figure 3) since a sphere may be formed from a combination of holomorphic double helices, which are fundamental eigenfunctions of the entropic Hamiltonian.

Parker & Jeynes 2019 [17] already showed that the double helix is a special case of the double logarithmic spiral, which models spiral galaxies (like our own Milky Way) remarkably well: that is, the QGT entropic description of spiral galaxies is an alternative explanation for their manifest stability and their double-armed logarithmic structure, an explanation that needs no unobserved (and apparently unobservable) entities.

Rather, the presence of a supermassive blackhole (SMBH) at the galactic centre, an object exhibiting the largest entropy (by far) as compared to any other object within the galaxy, itself becomes the (thermodynamic) explanation for the galactic spiral geometry and stability. That is also to say, the stupendously gigantic blackhole entropy of the SMBH dominates the entire galaxy and is certainly not “hidden” behind the SMBH’s event horizon.

With respect to the entropic Lagrangian–Hamiltonian canonical equations of state, some important fundamental relations were derived by Parker & Jeynes 2021 [44], in particular, the *entropic* Liouville equation in terms of the thermodynamic conjugate variables of the *entropic* Hamiltonian (the “hyperbolic position” and “entropic momentum”), which they used to derive not only the holographic principle, the Bekenstein–Hawking equation, and the *entropic* Uncertainty Principle (based on the Boltzmann constant), but also an interesting isomorphism between the Schrödinger equation and the *entropic* Partition Function.

Parker & Jeynes 2021 [45] also proved that *entropy production* (the complexified counterpart to energy) is a conserved quantity, as well as calculating the entropy production of black holes (a calculation confirmed subsequently by Parker & Jeynes 2023 [25]). However, note that the important derivations by Reguera et al. [31] of entropy production (leading to their Equations (12) and (22)) do not use the variational principles (maximum entropy, least action, etc.) in any way, whereas these principles (expressed in the relevant Euler–Lagrange equations) are central in the QGT formalism. We have already commented that Reguera et al. [31] very helpfully summarise how to recover discrete behaviours (characterised by the Fokker–Planck equation, see their Equation (57)) using canonical methods. But Parker & Jeynes 2023 [25]) also effectively recover the same features, since the diffusion equation (their Equation (1a)) may be regarded as a pared-down Fokker–Planck equation (without its drift coefficient). Moreover, the QGT treatment makes clear that the conservation laws apply to both *energy* and *entropy production*.

The point here is that all these calculations are on the basis that the alpha particle (in its ground state) is properly an atom—a unitary entity (than which nothing simpler exists). Another way of speaking about unitary entities is to say they have no parts! Now of course, the alpha particle consists of two protons and two neutrons so it seems wrong to say that “it has no parts”. However, its binding energy is very high (>7 MeV/nucleon) so even though it is not strictly “indivisible”, it is practically “indiscerpible” (that is, remarkably hard to “tear asunder”): it normally behaves as a “unitary entity” at the length scale of interest (which for the alpha is the diameter of a proton).

Of course, thermodynamics is normally about processes in which things are happening (where entropy is being created). QGT was initially about geometrical effects which applied to *stable* things (spiral galaxies, DNA, Buckminsterfullerene, the alpha particle, etc.) but Parker & Jeynes (2023) [46] have also used it to correctly calculate the lifetime of the free neutron, which is *unstable*: it decays to a proton, an electron, and an antineutrino (with a lifetime of about ¼ h). They also use QGT to correctly calculate the half-lives of other nuclei (including ^6^He, ^8^He, and ^8^Be). In these calculations, the Noether-conservation of entropy production (proved by PJ21b [45]) is the key fundamental principle being exploited.

It is important to emphasise that, even though an entity may be unitary, it is not necessarily stable (using the QGT formalism). That is to say, for the duration that it may exist, the unitary entity has the qualities of an “atom”; however, it may also have a half-life and spontaneously cease to exist as it fragments into constituent components. Thus while it exists, the atomic entity may be “unitary”; but once it fragments into sub-components (increasing entropy according to the Second Law), its ontology changes (since it no longer exists as unitary).

The ab initio calculations discussed above closely reproduce the observed values, showing that QGT is about *processes* as well as pure geometry. But geometrical methods in physics are now becoming widespread: Jeynes (2023) [47] has reviewed the use of comparable new methods in optics. Hestenes (2002) [16] has long argued for the systematic use of powerful geometrical algebra methods in physics, pointing out their ability to simplify complex arguments and make new understandings available. Physicists are starting to respond. For example, Aslani (2025) [28] has used the new methods to reanalyse the damped harmonic oscillator, obtaining more detailed results (and more easily) than those obtained with standard treatments.

QGT is scale-invariant. That is to say, it operates at all length scales; yet, the scale length always needs to be specified (as a frame of reference) in order to satisfy the principle of scale relativity (Auffray & Nottale 2008) [48]. It is this scale invariance that has facilitated the resolution of long-standing logical (“Bertrand”) paradoxes (Parker & Jeynes 2023) [49]. Another long-standing puzzle is the relation between general relativity and quantum mechanics: the one is important at cosmic scales, the other at the nanoscale; but the two accounts are mutually inconsistent even though both are extraordinarily accurate. We expect the scale-invariance of QGT to be crucial in addressing this.

Thermodynamics is often represented as being reducible to statistical mechanics, involving probabilities; and probabilities are also central in the discussion of entropy in modern information theories. But QGT has been used to show that *probability* is essentially a physical (not a mathematical) idea, with the sum rule for probabilities in general being hyperbolic (Parker & Jeynes 2025) [50] and thereby conforming to the scale-invariance of QGT.

As we already mentioned, in a far-reaching work Parker & Jeynes 2023 [25] have shown that reversible and irreversible processes can be treated entirely commensurately. Specifically, they flesh out the isomorphism already shown in 2019 [17] between physical relations expressed in *energy* terms and those expressed in *entropy* terms. Using a fully complexified treatment (including a complexification of time), they show that the Wick-rotated complex conjugate of the entropic Hamiltonian is just the *entropy production* (in holographic natural units). See Figure 4.

This makes it clear that *time* should be represented as a complex number: it was always clearly a simplification to think that it could be properly represented by a scalar. But taking this seriously, Parker et al. (2025) [39] have shown that a valid Lagrangian can be constructed entirely in complex time (with no spatial component), leading them to call it a “purposive Lagrangian” specifying the “entropic purpose” of the system (see Figure 5).

“Entropic purpose” is *defined* by Parker et al. as a line integral over the “purposive Lagrangian” of the system as it traverses a trajectory across the complex temporal plane (see Figure 5), comparable to action which is also defined as a line integral over an appropriate (spatial) Lagrangian. Entropic purpose also displays the same variational behaviour as action. That is, just as there is a principle of least action, so also (for the same reason) there is a principle of least entropic purpose.

Parker et al. also go on to prove that this quantity of *entropic purpose* is represented by the Shannon information *created* by the system as it navigates a temporal trajectory through complex time. This latter property touches what we could properly think of as the *purpose* of the system, since what the system *does* represents its purposes as it moves forward in time.

One could say that the power of QGT depends on the fact that its basic objects are holomorphic. The Riemann–Silberstein representation [26] of the free-space electromagnetic field is holomorphic [51], and photons (as the quanta of the EM field) are therefore unitary entities (indivisible *per se*). The EM field is a different thing from either of the electric or magnetic fields that compose it when they are bound together holomorphically.

In just the same way, the DNA molecule, when its two RNA halves are bound together holomorphically, is a different thing from its components: being holomorphic, DNA behaves as a unitary entity. Similarly, the neutron halos of both the ^6^He and ^8^He nuclei, being holomorphic, are also unitary—as are the pair of logarithmic arms in a spiral galaxy. In all these cases, the entities formed are *indiscerpible* (not “indivisible”); in particular they have a half-life, and Tour et al., 2025 [52] have discussed the case of polymer chains (like RNA). Unexpectedly, unitary entities prove to be remarkably flexible in their ontology!

## 4. QGT Formalism

Quantitative Geometrical Thermodynamics (QGT) was introduced in 2019 by Parker & Jeynes (PJ19 [17]), initially considering only stable things (like DNA). It is remarkable that a very close relation between entropy and information was earlier shown to explain the chirality of natural DNA (by Parker & Walker 2010 [40]). This is treated rigorously in Appendix A of PJ19 ([17]) using geometrical algebra methods (as hinted in Figure 2).

The QGT formalism was summarised formally in Section 2 of Parker et al., 2025 [39]: here we will sketch it more informally. Equation (C23) in Appendix C of PJ19 [17] gives the entropic Lagrangian *L*_S_ for the double-armed logarithmic spiral in hyperbolic 3D space *q*:(1)$$L_{S}\left(q,q^{\prime },x_{3}\right)=3m_{S}+\sum_{n=1}^{3}m_{S}\ln q_{n}^{\prime }-V_{S}\left(q\right)\;\;\;\;\;\;\;\;\;\;\;\;\;\;n=\left\{1,2,3\right\}$$
where *q_n_* represents the (hyperbolic) coordinate system (that is, the hyperbolic position); and *q*′ represents the “entropic momentum” (that is, an appropriate differential of *q*). For hyperbolic position and entropic momentum, see Equation (9) of PJ19 [17].

In particular, *q*′ ≡ d*q*/d*x*_3_, where in this entropic representation *x*_3_ is the (Euclidean) spatial coordinate associated with *q*_3_ (see Equation (B24) in Appendix B of PJ19 [17]) that would classically be the time dimension in the kinematic representation. This makes explicit the *geometrical* flavour of QGT: PJ19 [17] addresses the statics, not the dynamics. Stable structures have symmetries, and in QGT the general system (in space–time) is integrated in a way that makes the *x*_3_ symmetry axis behave mathematically as the time axis in standard kinematics (hence, we have d/d*x*_3_ instead of d/d*t*: see Table 1 of PJ19 for an explicit summary of the isomorphisms between the entropic and kinematic representations).

In Equation (1), *m*_S_ ≡ iκ_0_*k*_B_ is the *entropic mass*, where *k_B_* is Boltzmann’s constant and κ_0_ is a parameter of the representation (see PJ19 [17] Equation (7)) with the dimensions of wavenumber (that is, inverse length) and which determines the length scale of the system under consideration. *V_S_* is the potential entropy field (see Equation (C24) of Appendix C of PJ19 [17]).

*Entropic mass* is isomorphic to *inertial mass*, which has always been mysterious. However, the quantity in QGT defined as “entropic mass” acts in exactly the same way that inertial mass acts in the canonical kinematics equations, which is a surprise since QGT was not “designed” to exhibit an “entropic mass” quantity *per se*. It just emerged that way. Note that the “entropic mass” is found to be “imaginary” (i^2^ ≡ −1). However, what is particularly interesting is that the entropic mass is geometric in character, fundamentally based on the wavenumber (inverse wavelength) of the unitary structure under consideration. The entropic mass is perhaps simply the “imaginary” counterpart of inertial mass, in the same way that “imaginary energy” is equivalent to entropy production (see below: Equation (12) *passim*). Thus, it could well be that “entropic mass” is giving us a very important insight into how inertial mass is also to be understood from a geometrical and entropic perspective.

Using the Legendre transformation yields the entropic Hamiltonian *H*_S_, given by Equation (11) of PJ19 [17]:(2)$$H_{S}=\sum_{n=1}^{3}\left\{p_{n}q_{n}^{\prime }\right\}-L_{S}\;\;\;\;\;\;\;\;\;\;\;\;\;\;n=\left\{1,2,3\right\}$$
where as usual *p_n_* and *q_n_* are the (entropic) conjugate variables (“entropic momentum” and “hyperbolic position”) of QGT, where it is also correct to regard *q* and *q*′ as conjugate variables. Note that the entropic Hamiltonian is the sum of the kinetic entropy *T*_S_ and the potential entropy (*H*_S_ = *T*_S_ + *V*_S_) just as the (kinematical) Hamiltonian is the sum of the kinetic and potential energies.

As already previously highlighted, Parker & Jeynes 2019 [17] show that the double helix and the double logarithmic spiral are both holomorphic fundamental eigenfunctions of the entropic Hamiltonian (in hyperbolic space) since they satisfy the variational Euler–Lagrange equations indicating that they are maximum-entropy structures, thereby also proving the validity of the entropic analogue of the principle of least action (namely, the *principle of least exertion*). Note that Parker & Jeynes 2023 [42] also show that exertion is proportional to the quantity “caliber” defined by Edwin Jaynes [53], the exceptional usefulness of which is reviewed by Dixit et al. [54]

Parker & Jeynes 2020 [43] elegantly account for the stability of Buckminsterfullerene (C_60_) by showing that a sphere is exactly represented by a combination of a pair of double helices (see Figure 3), and therefore that the sphere is also a maximum-entropy geometry: in particular, “*the representation* [Σ] *of the spherical topology of the C_60_ molecule is mathematically the superposition of two holomorphic double-helices* [Σ = Σ*_a_* + Σ*_b_*], *and is therefore also itself holomorphic*”. The relevant Equation (6) of [43] are(3a)$$\Sigma_{a}=\frac{1}{2}e^{i\varphi }\left\{Re^{i\theta }{\underline{x}}_{1}-iRe^{i\theta }{\underline{x}}_{2}-R\left(1+i\right){\underline{x}}_{3}\right\}$$(3b)$$\Sigma_{b}=\frac{1}{2}e^{-i\varphi }\left\{Re^{i\theta }{\underline{x}}_{1}-iRe^{i\theta }{\underline{x}}_{2}+R\left(1+i\right){\underline{x}}_{3}\right\}$$
where *R* is the radius of the sphere and {*φ*, *θ*} are phases of the double helix, with {*x_n_*} *n* = 1, 2, 3 being the unit axes in Euclidean space. Because of its high symmetry, Buckminsterfullerene is easy to treat analytically; numerical methods will probably be needed to treat other fullerenes.

Parker & Jeynes have derived the entropic Liouville equation (Equation (3) of PJ21a [45]):(4)$$\frac{d\rho_{S}}{dx_{3}}=\frac{\partial \rho_{S}}{\partial x_{3}}-\sum_{n=1}^{N}\left(\frac{\partial H_{S}}{\partial q_{n}}\frac{\partial }{\partial p_{n}}-\frac{\partial H_{S}}{\partial p_{n}}\frac{\partial }{\partial q_{n}}\right)\rho_{S}=0$$
in terms of the entropic Hamiltonian *H_S_* and the density of states *ρ_S_*. This has very far-reaching consequences: they obtain from it not only the holographic principle and the Bekenstein–Hawking equation for the entropy of a MaxEnt entity (such as a black hole) but also the entropic Uncertainty Principle, represented by the quantum entropic canonical commutation relation (Equation (18a) of [45]):(5)$$\left[{\hat{q}}_{S},{\hat{p}}_{S}\right]=2k_{B}$$
where {${\hat{q}}_{S}$, ${\hat{p}}_{S}$} are the quantum operators (see Equation (18b) of [45]) derived from the entropic conjugate (generalised position and momentum) variables {*q*, *p*}, and *k_B_* is the Boltzmann constant as usual. The factor “2” in Equation (5) arises from the fact that the Boltzmann constant as defined is associated with the “amplitude” description, whereas the Planck constant has been defined with respect to the “amplitude squared” description for quantum mechanical quantities (such as probability: see further Parker & Jeynes 2023 [25] and Córdoba et al., 2013 [55]). Parker & Jeynes also demonstrate that a probability term *P_n_* of the entropic Partition Function *Z_S_* is isomorphic to a solution of the Schrödinger equation (Equation (16) of [45]):(6a)$$Z_{S}\equiv \sum_{n}e^{-\beta_{S}S_{n}}$$(6b)$$P_{n}\equiv \frac{e^{-\beta_{S}S_{n}}}{Z_{S}}$$
where *β_S_* ≡ 1/*k_B_* is the entropic equivalent to the conventional inverse temperature parameter *β* ≡ 1/*k_B_T*. Note that in the QGT formalism, the temperature *T* does not appear: *T* is a parameter that transforms between the kinematic and entropic domains (kinematic ↔ entropic: see also Table 1 of PJ19 [17]).

Since Equations (6a) and (6b) show that the Partition Function may be derived from QGT, they also mean that thermodynamics is *not* reducible to statistical mechanics (contradicting what is widely thought), a conclusion supported by Michael te Vrügt (2022) [56] in his careful discussion of coarse-graining in the context of the Mori–Zwanzig projection operator formalism. That is to say, thermodynamics is fundamental.

Parker & Jeynes “*[extend] QGT to generalise Onsager’s differo-integral formalism for the entropy production of systems in hyperbolic (Minkowski) 4-space [showing] that the relativistic entropic Hamiltonian directly determines the entropy production*”, also showing that “*both the entropic Hamiltonian and the entropy production are conserved quantities*” (PJ21b, [12]). They apply their formalism to heavily idealised spiral galaxies, represented only by their central supermassive black hole (SMBH) and ignoring all the stellar mass on the grounds that the virial mass of such galaxies is dominated by (unobservable) “dark matter”. This builds on the work of PJ19 [17] who have already shown that the virial mass of such (heavily idealised) spiral galaxies may be reasonably calculated by QGT, using the central SMBH entropy and certain simplifying assumptions. Thus, the effects of the supposed “dark matter” may instead be considered as a purely entropic phenomenon due to the (stupendously gigantic) central SMBH entropy. That is, the *stability and geometry* of the spiral galaxy is entailed by its central supermassive black hole.

What is interesting is that the entropy production $\dot{S}$−Φ of the idealised galaxy (that is, the black hole with mass *M*_BH_ and radius *r*_BH_), has *two* components {$\dot{S}$,Φ}, where one is the *entropic mass* (Equation (31) of [12]):(7)$$\Phi =c\left|m_{S}\right|=2\pi \sqrt{\frac{c^{5}}{\hbar G}}k_{B}$$
where *G* is the gravitational constant; ℏ is the reduced Planck constant; and as usual, *c* is the speed of light; and where for the Milky Way, Φ is 1.6 × 10^21^ J/K·s.

The other component is the *entropic momentum* (Equation (25) of [12]):(8)$$\dot{S}=cp=\frac{ck_{B}}{r_{\mathrm{BH}}}=\frac{c^{3}k_{B}}{2GM_{\mathrm{BH}}}$$
where for the Milky Way, $\dot{S}$ is 3.2 × 10^−25^ J/K·s.

The entropic mass (the volume-integrated dissipation function Φ, Equation (7)) is 46 orders of magnitude larger than the entropic momentum ($\dot{S}$, Equation (8)). Note that $\dot{S}$ is functionally similar to the Hawking radiation (only differing by a factor of ~100: compare Equations (27) and (28) of PJ21b [12]). Black holes cannot evaporate! Their Hawking radiation is tiny compared to the rate at which they must accrete mass.

But what is also interesting is that PJ21b [12] asserts an “*intimate relationship between the geometric structure of a spiral galaxy and the dissipative processes intrinsic to it*” also describing spiral galaxies as “*Maximum-Entropy unitary objects obeying the Principle of Least Exertion*”: the variational principles pervade all of our closest descriptions of reality. Black holes are archetypal unitary entities, as indeed are also alpha particles. “*Both supermassive black holes and alpha particles are represented in QGT as unitary entities requiring only four scalar quantities for a complete specification: the mass, the charge, the spin, and a scaling parameter*” ([12], citing a preprint of [11]). However, whereas the alpha particle is stable with zero entropy production, a black hole is highly dissipative and has a high (positive) entropy production.

Thus, where alphas embody the double helix, black holes embody the double logarithmic spiral (of which the double helix is but a special case): the double logarithmic spiral shape signals dissipation (non-zero entropy production) in contrast to the double helix signalling no dissipation (zero entropy production). Where alphas are trivially time-reversible (being stable: unchanging in time), black holes definitely are not! (Note that, like the alpha, the photon also has zero entropy production and is trivially time-reversible.) However, both the alpha particle and a black hole are unitary entities; they both obey the holographic principle, with their entropy *S* proportional to their surface area *A*:(9)$$S=\frac{A}{\lambda^{2}}k_{B}=4\pi \frac{r^{2}}{\lambda^{2}}k_{B}$$
where *λ* defines the length scale, and (since such unitary entities are necessarily spherical) *A* ≡ 4*π**r*^2^ where *r* is the entity’s characteristic radius. A black hole’s characteristic radius is its Schwarzschild radius *r* ≡ *r*_BH_, and its length scale is the Planck length *λ* ≡ *l_P_*. In contrast, the radius of the alpha particle is *r* ≡ *r_α_* = 1.63 fm, and its length scale is determined by the proton diameter, *λ* ≡ 1.682 fm (as observed by muon scattering [57] supported by the Jefferson Laboratory “*proton charge radius experiment*” [58]).

Parker et al., 2022 ([11], and see Figure 1) use the formalism of PJ19 [17] and the results of PJ21a [45] to calculate the matter radius of the alpha particle (and other nuclei) ab initio from QGT, using no quantum mechanics at all but treating the alpha as a *unitary entity* (which has no parts). In particular, their treatment rests on a generalisation of the Bekenstein–Hawking equation and on the holographic principle, both of which are derived (by PJ21a [45]) entirely in a QGT formalism.

To this point, QGT has been articulating the static properties of maximum-entropy (MaxEnt) entities. (Note that black holes are MaxEnt even though their entropy production is non-zero.) But in 2023, Parker & Jeynes [46] used QGT to treat the *process* of beta decay for various nuclei. In particular, the entropy production Π is related to the exponential-decay time constant *τ* of a process together with the change in the number of degrees of freedom Δ of the system (Equation (2) of [46]):(10)$$\Pi =\frac{e^{-\Delta }}{\tau }S_{0}$$
where the appropriate relation for the change in entropy production ΔΠ is isomorphic to the Planck–Einstein relation Δ*E* = *hf* (Equation (1) of [46]):(11)$$\Delta \Pi =ck_{B}/\lambda$$
where Equation (11) again invokes the (entropic) length scale *λ* associated with the phenomenon being described, such that *c* = *fλ* as usual. Again, we see the presence of the Boltzmann constant *k_B_* in Equation (11) as the entropic analogue to the Planck constant for kinematics.

What is astonishing is that Parker & Jeynes (see Equation (23c) of PJ23a [25]) have shown explicitly that energy and entropy production really are two sides of the same coin, specifically that the (complex) Hamiltonian *H_z_* and the (complex) entropy production Π*_z_* are essentially Hilbert transforms of each other:(12)$$iH_{z}^{*}=\frac{h}{4\pi k_{B}}\Pi_{z}$$
that is, the (Wick-rotated) *complex conjugate* of the (complexified) “entropic Hamiltonian” simply equals the (complexified) entropy production (in holographic natural units—that is, over the whole 4π sphere: see Figure 4). What is required here (to apply the Hilbert transforms and make use of the power of complex analysis) is that the whole treatment is complexified including, crucially, time itself. This fundamental identity between the *complex-valued* Hamiltonian and the *complex-valued* entropy production is yielded by analytical continuation into the complex frequency plane and using the Cauchy–Riemann relations. 

Equation (12) is extended surprisingly by Parker et al., 2025 (PW25a: Equation (8) of [39]), who are able to define a “purposive Lagrangian” *L_p_* entirely in complex time (with no spatial component):(13)$$L_{P}=cm_{S}\left(\ln q_{t}^{\prime }+\ln q_{\tau }^{\prime }\right)+2cm_{S}\left[\Lambda T-\ln\left(1-\Lambda T\right)\right]+cm_{S}\ln K_{t}K_{\tau }$$
where *K* are entropic constants, *T* = |−τ + i*t*| is the empirical (measured) time, and *q’* are the differentials with respect to the empirical (essentially, Euclidean and local time) *T* associated with hyperbolic time *q* (see Equation (7) of PW25a [39]): τ is the “real” part and *t* is the “imaginary” part of complex Euclidean time, although both τ and *t* are just as real (or imaginary!) as each other. Equation (13) is formally similar to the equation for the Lagrangian of a dissipative and entropy-producing logarithmic double spiral, where Λ represents its logarithmically varying radius (see Equation (19) of [17]). But here the formalism (Equation (13)) is defined in the (hyperbolic) 2-D *q*-space given by the complex time plane.

Parker et al. [39] prove that the purposive Lagrangian of Equation (13) is valid by demonstrating that it satisfies the Euler–Lagrange equation describing the appropriate variational principle, which they call the *principle of least (entropic) purpose*. And “*entropic purpose*” itself is defined as a line integral on the purposive Lagrangian (just as *action* is also defined as a line integral on an appropriate Lagrangian: see Figure 5). Moreover, Parker et al. prove that the *entropic purpose* of a process is measured by the *Shannon information* created by that process.

Finally, Parker et al., 2025 (PW25b: [50]) explicitly address the issue of *recursion*, which involves a hyperbolic mathematical object; again, this is strongly resonant of QGT quantities. Recursion is deeply relevant to any discussion of the probabilities of real processes (especially in living beings). The present conventional treatment of *probability* is specialised to simple applications (games of chance) which do not admit recursion. But Parker et al. (see Equation (12) of PW25b [50]) have generalised this treatment of the probabilistic or function to give a new (hyperbolic) formula for the probability *p* of {(*A* or *B*) given *C*}:(14)$$p\left(A\;\mathrm{OR}\;B|C\right)=\frac{p\left(A|C\right)+p\left(B|C\right)}{1+p\left(A|C\right)p\left(B|C\right)}$$

The mathematical similarity of Equation (14) to how kinematic velocities in special relativity are added is clearly apparent. This hyperbolic sum rule is highly relevant to situations where the probability must be estimated of multiple (possibly recursive) hypotheses under multiple (possibly recursive) conditionalities. This includes most real situations! The usual treatment of probabilities today only treats simple (non-recursive) cases in which there are only a limited (small) number of both hypotheses and conditionalities. But real systems typically have a large number of these, and are also normally heavily recursive, e.g., as seen in the large language models (LLMs) of artificial intelligence (AI). This is clear from Philip Ball’s description of “*How Life Works*” (2023) [59], and it is also in view in Terrence Deacon’s treatment of “*Incomplete Nature*” (2011) [60].

## 5. Atoms

What we now usually call “atoms” (the elements of the Periodic Table) are not properly atoms at all, since they are all so easy to ionise (with first ionisation energies typically being a few eV, they are hardly “uncuttable”)! Parker et al., 2022 ([11], building on their independent derivation of the Bekenstein–Hawking equation from the entropic Liouville Equation: see Equation (12a) of PJ21a [45]) have shown that the alpha particle is a “unitary entity” (since we may regard a MaxEnt object obeying the holographic principle [61] as *unitary*); that is, in its ground state it can be considered to “have no parts”. This is remarkable since we know that it consists of four nucleons: it is the enormous binding energy of >7 MeV per nucleon that makes it *indiscerpible* (resisting being “torn asunder”). Schlesinger in 1961 [62] elegantly made a philosophical case against the (widely assumed but false) idea that “smaller” is necessarily “more fundamental”; here, Parker and others have shown a physical case making the same point.

But we know that the Bekenstein–Hawking equation was originally derived for black holes, so that what is true for alpha particles is also true for black holes (BHs): BHs are “unitary entities” too! But where the alpha is *indiscerpible*, the BH truly is *indivisible*: the archetypal atom.

Parker & Jeynes have also calculated two components of the entropy production of BHs (Equations (7) and (8): see PJ21b [12], confirmed by PJ23a [25]): the tiny (negative) “Hawking radiation” of BHs is swamped by a much larger (positive) component so that they are never expected to evaporate, however slowly, although it is still an open question how the supermassive BHs found at the centres of galaxies reached their observed sizes.

Thus, QGT methods have revealed that both alpha particles and black holes are real atoms, that is, “uncuttable” entities, either absolutely or practically (*indivisibly* or *indiscerpibly*). What is critical in these calculations is the characteristic *scale* of the entity: for the alpha this is the proton diameter (a measurable quantity), and for the BH it is the Planck length. The enormous disparity in size between the alpha diameter (3.3 fm) and the BH diameter (52 × 10^6^ km for the SMBH at the centre of the Milky Way) is due to the enormous mass of the BH (which determines the size of the event horizon), and their vastly different length scales. This use of the length scale is in effect our generalisation of the holographic principle.

Building on the idea (central in QGT) of *holomorphism*, we have asserted that holomorphic entities, being MaxEnt and also obeying the holographic principle, are *per se* unitary. Atoms are, properly, holomorphic (they have a “shape of wholeness”). Table 2 of Parker et al., 2022 [11] shows the nuclear sizes of the “helium series” (^4^He, ^8^Be, ^12^C, ^16^O, ^20^Ne, ^24^Mg, ^28^Si, ^32^S, ^36^Ar, and ^40^Ca) all correctly calculated ab initio from QGT. These nuclei are all sums of alpha particles (that is, they conform to the holomorphic addition of unitary entities), so that they too may also be regarded as unitary. But experimentally, such nuclei are also indiscerpible (while noting that their identity depends on their unity—torn asunder, they cease to exist *per se*). In the same way, the 17th century philosophers regarded *poems* as “atoms” (see [1]); since the very existence of a poem depends on its integrity. In our terms those philosophers were right, since poems are, properly, indiscerpible.

Disregarding its encoded genetic information, we have shown ([17]) how the DNA molecule is also unitary (being holomorphic, MaxEnt, and obeying the holographic principle). The Buckminsterfullerene molecule (C_60_, [43]), being a sum of holomorphic entities is also unitary in the same way that the “helium series” is.

Note that every unitary entity (such as an alpha particle or a BH) is contingent: it may be created or cease to exist. Alphas are created in the Sun, releasing energy; this process may also be reversed, given sufficient energy. Also, we suspect that electrons (for example) are “fundamental” (that is, unitary); but if an electron meets a positron it too will be annihilated—transforming into two photons (511 keV gammas), noting that in our terms photons are also unitary (being represented by the holomorphic Riemann–Silberstein vector). We have previously discussed the lifetimes of other entities [46]. We leave the resulting ontological puzzles to the philosophers.

## 6. Discussion

“Shannon information” is an engineering concept used for developing and building the photonic communications networks that underpin the information revolution of today. Telecoms engineers use the information carried by the network as a syntactical object. That is, the engineers explicitly *exclude* consideration of the semantic content of the “information”: they are deliberately uninterested in the *meanings* of the messages carried by the networks. Similarly, “entropic purpose” is a physical idea that specifically *excludes* the human properties of *purposes*: Aristotelian teleology was excluded from physics in the seventeenth century—and a very good thing too!

Note that just as *action* is defined as a line integral on the appropriate Lagrangian, *entropic purpose* is defined as a line integral on the “*purposive Lagrangian*” (given in Equation (13)). The (somewhat provocative) names are justified precisely because Parker et al. also prove that the “entropic purpose” of a system may be measured by the new Shannon information generated by the system [39].

But in Figure 5, Aristotle is considering a Nautilus shell. We know that Aristotle knew about Nautilus since it is mentioned in his “Τῶν περὶ τὰ ζῷα ἱστοριῶν” (known in Europe as “*Historia Animalium*”, 4th century BCE), as noted by von Lieven & Humar (2008) [63]. It seems that there is *something* to be said for Aristotelian teleology, however wrong it may be, so that Carlo Rovelli (2014, p. 27) [64] is right to say of Aristotle’s physics, “*it’s not at all bad*”. The ancients liked Nautilus because it was an exemplar of the Golden Ratio (see Figure 9 p. 33 of Charles Seife, 2000 [65]). But Nautilus is also a rather perfect physical exemplar of the *double logarithmic spiral*, which Parker & Jeynes, 2019 [17], have demonstrated to be a fundamental eigenfunction of the entropic Hamiltonian.

We are not here rehabilitating Aristotelian teleology. But everyone knows that *purposes* are a characteristic of living things, and physicists claiming that these “purposes” are merely inconsequential epiphenomena are, quite properly, ignored by sensible people. We are here claiming that a correct physics that gives due weight to the Second Law of Thermodynamics does recognise “*entropic purpose*”, which is a cut-down version of “*purpose*”—that is, shorn of its human value just as Shannon information is.

We should note that the classical “Loschmidt Paradox” that has plagued statistical mechanics ever since Boltzmann formulated it has been resolved by QGT; see Parker & Jeynes 2023 [25] who show how to treat reversibility and irreversibility commensurately: the irreversible cases are seamlessly included in the Lagrangian–Hamiltonian articulation of the Second Law (axiomatic for QGT). The crux of this QGT treatment is the understanding that the treatment must be fully complexified—that is, *time* must also be represented as a complex number. Really, this is not very surprising since everyone knows that time is a slippery and very difficult philosophical concept. It is remarkable that the scientific community has got so far with the obviously crude approximation of time as representable by a simple scalar number.

We should point out that complex numbers underpin all modern physics, and are also at the root of the Lagrangian–Hamiltonian representation of classical physics. Of course, the reason is the extraordinary power of complex analysis, which Roger Penrose calls “magic” (2004 [66], ch.4; figuratively, of course)! Reality is complex. Complexified time (“5D spacekime”) is also used systematically by Ivo Dinov’s group to simplify Big Data problems (see for example Zhang et al., 2024 [67], or Niraula et al., 2025 [68]).

The physical significance of “real” and “imaginary” is highlighted by Equation (12): complex numbers are real! Geometric algebra should be used to represent reality: in this formalism “imaginary” numbers are most clearly isomorphic to the pseudoscalar, but the other higher-order vectors also have “imaginary” properties (see Appendix B of Aslani [28], and on geometric algebra see also Hestenes [16]). It is obvious that the representation of reality using only scalar numbers is a simplification. Reality is complex, and “imaginary” is merely a (historical and, unfortunately, misleading) label expressing certain geometrical properties of quantities when described in the appropriate physical space.

An intrinsic property of atoms is that they are necessarily discrete; but only a *non*-atomic description is consistent with the continuity required by complex analysis. We regard “atomism” to be paradoxical in the same way that nature has the discrete/continuous properties also seen in the wave–particle duality nature of quantum mechanics (and intrinsic to the Uncertainty Principle). But clearly there is still a formal problem regarding the description of a discrete reality using complex analysis. We regard this problem as out of our present scope (although we have glanced at it above in the context of the Fokker–Planck equation [31]).

We regard Penrose’s “*Road to Reality*” (2004) [66] as a seminal summary of physics but comment on certain surprising omissions. He mentions neither the concept of “information”, nor Carathéodory; and therefore his treatment of entropy is also somewhat inadequate, which is surprising in view of the space he gives to black holes and thermodynamics.

QGT does take the Second Law as axiomatic, following Einstein’s lead: he said, “*[thermodynamics] will never be overthrown*” [69]. It seems preferable to choose this axiomatic basis since, despite their outstanding observational accuracy, the currently popular alternatives (quantum mechanics and general relativity) are mutually incompatible: the former is necessarily non-local but the latter necessarily local [70]. Their reversibility does add elegance, but at the expense of credibility since every process we know of is irreversible; a circumstance guaranteed by the Second Law!

The equipartition theorem entails the Partition Function and the procedures used to establish the granularity of the system. In particular, the Markovian (“coarse-graining”) assumption that is used to derive the Partition Function can also be used to show how irreversibility may “emerge” from (reversible) classical mechanics (te Vrügt 2025) [71]. In this way at least, thermodynamics is fundamental (te Vrügt et al., 2022) [72]. This conclusion is also supported in a recent work by Kian Salimkhani [73] which shows how the (asymmetrical) arrow of time is consistent with the symmetry properties of quantum mechanics.

## 7. Conclusions

Unitary entities are the true atoms. That is, unitary entities are “fundamental” in important ways. “Atoms” were originally conceived as “uncuttable”, but of course what we now usually call “atoms” (comprising a nucleus and an electronic shell) are very easily ionised—that is, they are very far from being “uncuttable”. A “unitary entity” is “that than which exists nothing simpler”: that is, a unitary entity has, or may be considered as having, no parts (or, equivalently, it has the minimum number of degrees of freedom). “Atoms” are either actually “indivisible” or they may be practically “indivisible” (“*indiscerpible*” in Henry More’s 17th century terminology). Indivisible things are such that it is unreasonable to consider them divisible, where indiscerpible things are just very hard to divide.

We have shown that the alpha particle (in its ground state) may correctly be considered as a unitary entity [11]. This upsets the conventional view that regards smaller things as being “more fundamental” than composite things (such as the alpha, composed of four nucleons). Rather, the length scale adopted to analyse any phenomenon determines its “fundamentality”. The length scale is required for a correct unitary description, just as in microscopy the wavelength determines the resolution of what can be observed.

We have also asserted that black holes are unitary [39]. This is startling since black holes are enigmatic and extreme phenomena, not obviously “simple”. But unitary entities are such that “no simpler exists”, that is: they have no parts and the minimum number of degrees of freedom. Black holes are certainly indivisible, and having a spherical geometry they are an exemplar of a fundamental eigenfunction of the entropic Hamiltonian (which entails a non-zero entropy production in this case).

Michael te Vrügt et al. [72] have shown that thermodynamics is fundamental, and we have shown using thermodynamics methods how both black holes and alpha particles may be considered unitary entities or “atoms”: uncuttable entities that are either *indivisible* (BHs) or *indiscerpible* (alphas).

## Figures and Tables

**Figure 2 entropy-27-01119-f002:**
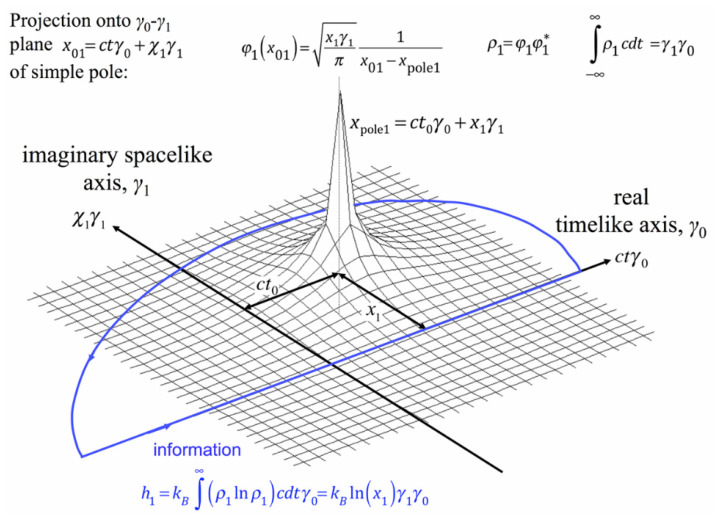
Projection of a simple pole onto the γ_0_ − γ_1_ space–time plane using the geometric algebra formalism, with resulting information calculated by integration along the time-like axis (reproduced from Parker & Jeynes, 2019 [17] Figure A.1; modified from Figure 1 of Parker & Walker 2010 [40]).

**Figure 3 entropy-27-01119-f003:**
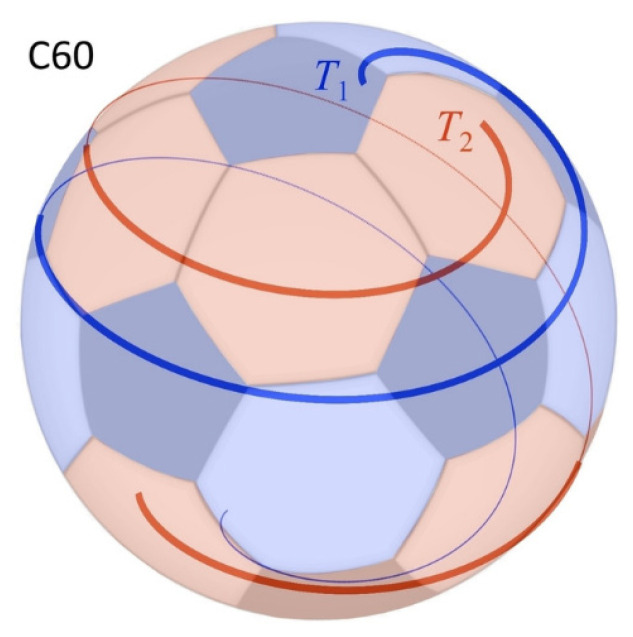
Representation of double spiral trajectories on the surface of a C60 molecule (bold lines: in front of sphere; thin lines: to the rear of the sphere). Reproduced from Figure 3 of Parker & Jeynes, *ChemistrySelect* **5** (2020) 5–11 [43].

**Figure 4 entropy-27-01119-f004:**
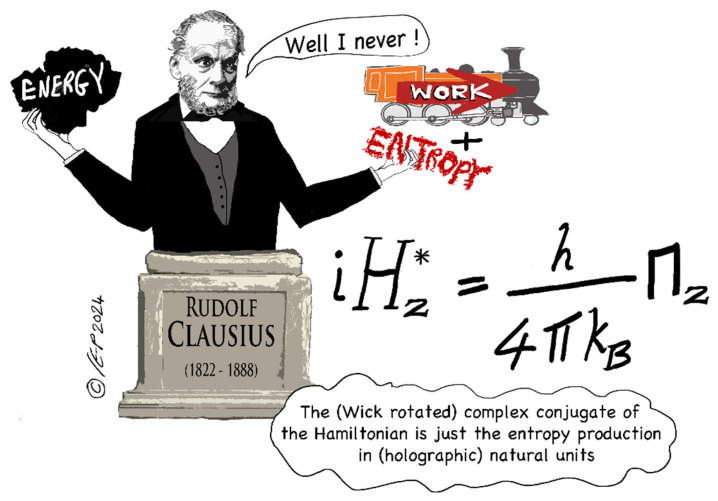
Graphical abstract of Parker & Jeynes 2023 [25]. Picture credit: Christine Evans-Pughe, www.howandwhy.com (accessed on 27 October 2025).

**Figure 5 entropy-27-01119-f005:**
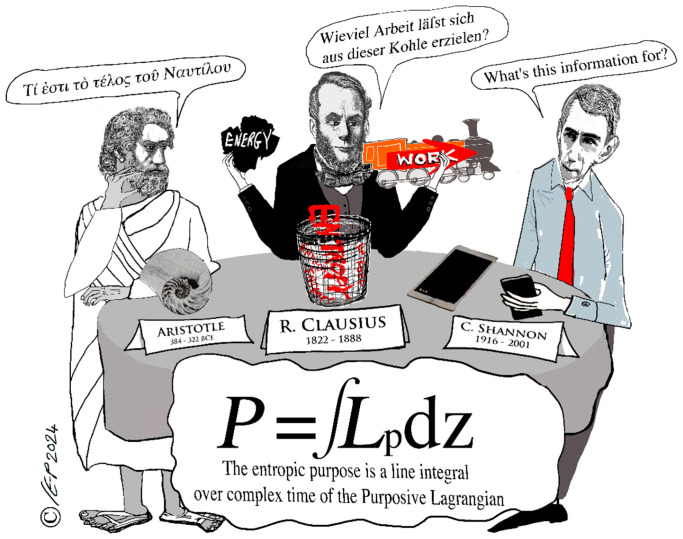
Graphical abstract of Parker et al., 2025 [39]. Picture credit: Christine Evans-Pughe, www.howandwhy.com (accessed on 27 October 2025). Aristotle is saying, “*What’s Nautilus’ purpose?*” and Clausius is saying, “*How much work is in this coal?*”.

## Data Availability

All data are in the article.

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
