# Peer review of "Unitary Entities Are the True “Atoms”"

_entropy, 2025, doi:10.3390/e27111119_

Round 1
Reviewer 1 Report
Comments and Suggestions for Authors
This manuscript is a review of Quantitative Geometrical Thermodynamics (QGT), extending the authors’ previous work and positioning “unitary entities” as the true “atoms” of physical science, with examples ranging from DNA and Buckminsterfullerene to alpha particles, neutron halos, and spiral galaxies. The paper is ambitious and provocative, bridging thermodynamics, geometry, and information theory. The manuscript is interesting and potentially impactful for Entropy's interdisciplinary readership, but it requires some minor revisions to ensure accessibility and rigor:
- In Section 3 (Holomorphism): The authors can improve the explanation of how holomorphic functions relate to physical stability; this will emphasize the novelty of this approach compared to conventional treatments.
- In Section 4 (Formalism): The authors can expand the explanation of the role of “entropic mass” and the meaning of “imaginary” quantities in physical terms.
- In Section 6 (Discussion): The authors should present claims about entropic purpose more cautiously; they should indicate which are formal mathematical results and which are interpretative analogies.
In general, some passages are long; breaking them into shorter paragraphs would improve flow.
Reviewer 2 Report
Comments and Suggestions for Authors
In the conceptual framework of the generally accepted Quantitative Geometrical Thermodynamics (QGT), several approaches can be considered, such as mesoscopic nonequilibrium thermodynamics (mNET), as seen in Reguera, D.; Rubi, J.M.; Vilar, J.M.G. J. Phys. Chem. B 2005, 109, 21502 9, a feature). The geometry of the mnet can perhaps be better envisaged when looking into a paper by the principal author https://www.sciencedirect.com/science/article/abs/pii/S0378437103002826,
which certain d-dimensional geometrical expositions are offered for a grains-containing multi-atomic system.
Moreover, the concept of monads as 'unitary atoms' as dated somehow back to Leibniz looks obsolete and hardly efficient in terms of the modern times of high-quality measurements, but looks intriguing as the hallmark of the properties of helium nuclei https://onlinelibrary.wiley.com/doi/epdf/10.1002/andp.202270003, as the authors of the review proudly announce.
In general, this review paper is quite intriguing, makes some sense from the historical perspective, but it ought to be more focused on the current physics nomenclature and its legitimate subject, not overriding too much the Principia, as Sir Isaak Newton would like to see it when still with us.
Let me propose a case-sensitive and diligent major revision, paragraph-by-paragraph of the text proposed for acceptance, according to what has generally been indicated above.
Round 2
Reviewer 2 Report
Comments and Suggestions for Authors
The revision of the manuscript by Jeynes and Parker on the Quantitative Geometrical Thermodynamics has been done correctly, and the answers to the reviewer's questions are fine, also provoking certain future discussions, which is, however, irrelevant to the present positive judgement. The usage of 12 out of 73 papers cited, thus, at a level of 16-17 percent of the whole, is appropriate.